# Calibration and Uncertainty Estimation Challenges in Self-Supervised Chest X-ray Pathology Classification Models

**Jenny Xu**                                                                JENNYXU6@STANFORD.EDU

*Stanford University*

**Pranav Rajpurkar**                                          PRANAV_RAJPURKAR@HMS.HARVARD.EDU

*Harvard University*

## Abstract

Uncertainty quantification is crucial for the safe deployment of AI systems in clinical radiology. We analyze the calibration of CheXzero (Tiu et al., 2022), a high-performance self-supervised model for chest X-ray pathology detection, on two external datasets and evaluate the effectiveness of two common uncertainty estimation methods: Maximum Softmax Probabilities (MSP) and Monte Carlo Dropout. Our analysis reveals poor calibration on both external datasets, with Expected Calibration Error (ECE) scores ranging from 0.12 to 0.41. Furthermore, we find that the model's prediction accuracy does not correlate with the uncertainty scores derived from MSP and Monte Carlo Dropout. These findings highlight the need for more robust uncertainty quantification methods to ensure the trustworthiness of AI-assisted clinical decision-making. [1]

**Keywords:** Uncertainty estimation, self-supervision, chest X-ray pathology classification

## 1. Introduction

Deep learning has revolutionized automated medical image analysis (Han et al., 2023; Rajpurkar and Lungren, 2023). However, safely integrating these AI systems into clinical radiology practice requires understanding their level of uncertainty when making predictions. Prior works have explored uncertainty estimation methods for clinical AI applications, including ensemble methods (Guo et al., 2022), Bayesian methods like Monte Carlo Dropout (Dohopolski et al., 2020), Dempster-Shafer Theory (Ghesu et al., 2021), and test-time augmentation (Dong et al., 2021).

Self-supervised learning enables models to learn from vast amounts of unlabeled data, which is critical for medical image interpretation. CheXzero, a state-of-the-art self-supervised method based on the Contrastive Language-Image Pre-Training (CLIP) model (Radford et al., 2021), achieves X-ray pathology classification accuracies comparable to radiologists (Tiu et al., 2022) on external datasets. While prior works have investigated the robustness of CLIP models on natural image classification tasks (Tu et al., 2024), studies have not evaluated the uncertainty estimation of CLIP-derived models for medical image assessment.

In this work, we focus on evaluating the calibration of CheXzero and assessing the application of two uncertainty quantification methods: maximum softmax probability (MSP) and Monte Carlo Dropout. Our findings reveal that CheXzero exhibits poor calibration when deployed on two external datasets. Furthermore, our preliminary analysis shows that the model's prediction accuracy does not directly correlate with the uncertainty scores derived from MSP or Monte Carlo Dropout. These results emphasize the importance of thoroughly evaluating model trustworthiness before deploying healthcare AI solutions and highlight the need for further research to ensure the safe assessment of model uncertainty.

---

1. Source code is available at https://github.com/jennyziyi-xu/CheXzero-Uncertainty-Estimation

## 2. Methods

**CheXzero Model** CheXzero (Tiu et al., 2022) is a self-supervised multi-label classification model for detecting pathologies in chest X-rays. In our experiments, we use the ensemble of top-performing checkpoints provided by (Tiu et al., 2022) without fine-tuning on the test datasets. CheXzero is based on the Contrastive Language-Image Pre-Training (CLIP) architecture. During inference, the model receives positive and negative prompts for each disease condition in order to output the softmax probabilities for the presence or absence of a condition.

**Datasets** We evaluate CheXzero on two datasets. We focus on five CheXpert competition pathologies: pleural effusion, atelectasis, cardiomegaly, consolidation, and edema.

- CheXpert: The entire CheXpert test set, consisting of 500 chest X-ray images labeled for the presence of 14 different conditions.
- PadChest: A dataset containing 160,868 chest X-ray images labeled with 174 different radiographic findings. We use a randomly sampled subset of 2,985 images in our experiments.

**Model Calibration Assessment** We assess model calibration using two methods:

- Expected Calibration Error (ECE): We use the softmax probability as the confidence score. $ECE = \sum_{m=1}^{M} \frac{|B_m|}{n} |acc(B_m) - conf(B_m)|$ measures the difference between the model's predicted probabilities and the observed probabilities (Naeini et al., 2015), where $conf(B_m)$ is the average confidence of bin $B_m$ and $acc(B_m)$ is the accuracy.
- Median Removal: We sort the softmax probabilities $p_i$ for the presence of each condition and iteratively remove 0-50% of samples with median probabilities while computing the AUROC score. If the model is well-calibrated, a higher softmax probability indicates higher certainty about the presence of a condition, while a lower probability indicates higher certainty about its absence.

**Uncertainty Estimation** We evaluate two uncertainty estimation methods:

- Maximum Softmax Probability (MSP): For each X-ray image, we define the certainty score as $s = \max_i p_i$, where $p_i$ is the softmax probability for the presence of condition $i$. We calculate the AUROC score after removing 0-50% of samples with the lowest certainty scores.
- Monte Carlo Dropout (Gal and Ghahramani, 2016): Using the pre-trained CheXzero model, we randomly prune 15% of the output weights in all MultiheadAttention layers and obtain predictions for all diseases. This process is repeated 30 times. For each condition and sample, we compute the standard deviation of the 30 $p_i$ values as the uncertainty score, where $p_i$ is the softmax probability for the presence of a condition. The uncertainty scores are sorted and we iteratively remove 0-50% of samples with the highest uncertainty scores.

## 3. Results

**Model Calibration Assessment**

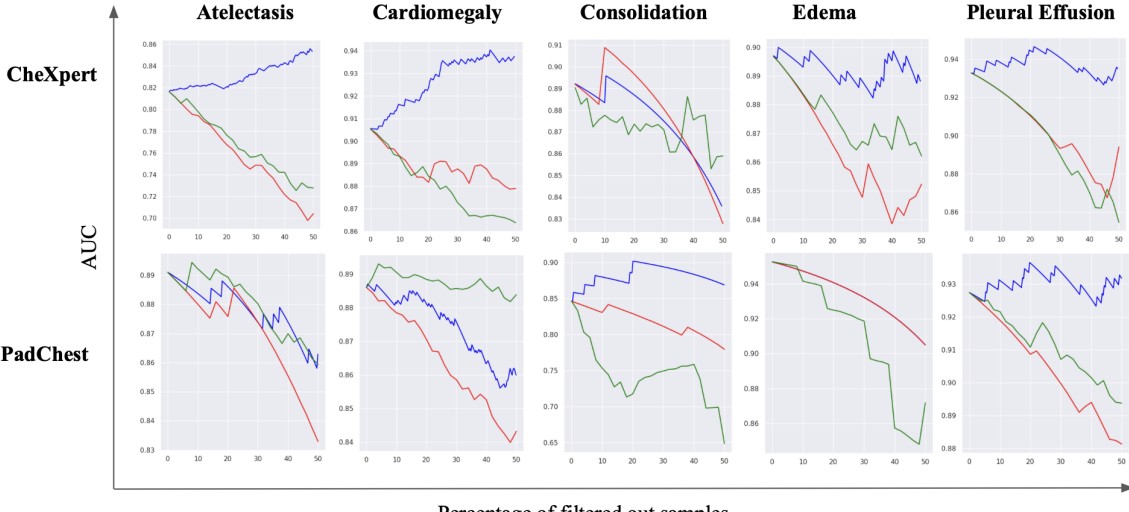

Figure 1: A comparison of Median Removal (blue), MSP (red) and Monte Carlo Dropout (green) on CheXpert and PadChest. Blue curve overlapping red curve for Edema.

- The ECE scores range from 0.12 to 0.35 on the CheXpert dataset and from 0.37 to 0.41 on the PadChest dataset for the five conditions. A well-calibrated model should have an ECE score close to 0.
- Median Removal: The plot of the percentage of removed samples against AUC should show an increasing trend for a well-calibrated model. On the CheXpert dataset, atelectasis and cardiomegaly show an increasing trend. None of the conditions show a clear increasing trend on the PadChest dataset (Fig.1). These findings suggest the model isn't well-calibrated on the external datasets. Fig. 1 shows that median removal outperforms both MSP and MCD uncertainty estimation methods.

**Uncertainty Estimation**

- MSP: As shown by the red curves in Fig. 1, AUC decreases across all five conditions when we filter out samples with low MSP scores, suggesting a prediction with high softmax probability does not necessarily correspond to low uncertainty.
- Monte Carlo Dropout: As shown by the green curve, AUC decreases when we filter out samples with high uncertainty scores derived from MCD's standard deviation. This indicates that MCD might not be a reliable uncertainty estimation measure here.
- The decreasing trends can be explained by the effect of filtering out true positives or true negatives from an original model with high AUC.

## 4. Conclusion

Our findings underscore the need to explore novel calibration and uncertainty estimation approaches, as well as establishing standardized evaluation frameworks for self-supervised medical image classification models.

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
