# OpenReview forum: "Calibration and Uncertainty Estimation Challenges in Self-Supervised Chest X-ray Pathology Classification Models"
_MIDL.io/2024/Short_Papers — MIDL 2024 Short Papers_

### Official Review · Reviewer_FcrK · 2024-04-24

**Confidence:** 5
**Final Rating:** 4

**Review:**

This study investigates the calibration performance of a state of the art self supervised method (CheXzero) for pathology detection in chest X-ray images. The influence of the calibration property is also investigated through the analysis of uncertainty predictions.

The strengths of this work are:
1) the relevance of the topic of this work
2) the relevance of the experiments and the metrics that have been chosen

The main weakness of this work is :
1) it would have been interesting to implement a recalibration method at the output of the CheXzero model, which would have made it possible to validate certain hypotheses formulated in the conclusions of the paper

Main feedback: This study provides interesting insights into the calibration requirements for the classification of pathologies using chest radiography.

---

### Decision · Program_Chairs · 2024-04-26

Accept